# Internalized Homonegativity and Sexual Quality of Life in Italian Lesbian and Bisexual Women

**DOI:** 10.3390/healthcare12060638

**Published:** 2024-03-12

**Authors:** Sofia Pavanello Decaro, Antonio Prunas

**Affiliations:** Department of Psychology, University of Milan-Bicocca, 20126 Milan, Italy; antonio.prunas@unimib.it

**Keywords:** sexual quality of life, internalized homonegativity, lesbian, bisexual women, minority stress model

## Abstract

This study investigates the relationship between sexual quality of life (SQoL) and internalized homonegativity among Italian lesbian and bisexual cisgender women, drawing on the minority stress model. The aim of this study is to compare levels of internalized homonegativity and SQoL between the two groups, exploring the association between these variables. We used a quantitative methodology based on a questionnaire. The data were collected through an online questionnaire from 686 women, including 217 lesbians and 469 bisexuals, using the Lesbian Internalized Homophobia Scale and the Female-Sexual Quality of Life Questionnaire. Statistical analyses, including t-tests and linear regression, were performed to assess group differences and predictors of SQoL. The findings support the hypothesis that bisexual women may experience higher levels of internalized homonegativity. Additionally, the study reveals disparities in SQoL, with lesbian women reporting better outcomes. The linear regression model confirmed a significant negative association between internalized homonegativity and SQoL. The results highlight the need for further research on factors influencing sexual well-being in sexual minority women, and the need to give thorough attention to specific sexual identities in clinical and research practice.

## 1. Introduction

Current sexological research and clinical practice extends beyond sexual functioning, in line with the World Health Organization’s definition, which labels sexual health as “a state of physical, emotional, mental and social well-being in relation to sexuality, not merely the absence of disease, dysfunction or infirmity” [1]. Sexual quality of life (SQoL) can be defined as a way of understanding the subjective experience of sexual health and has recently been used as an outcome variable in sexological research [2].

Previous literature has documented that there are stressors that can impact SQoL, and that these can vary based on a person’s sexual identity. For instance, people who belong to the LGBTQIA+ (lesbian, gay, bisexual, transgender/non-binary, queer, intersex, asexual/aromantic) community are at risk of being discriminated against and facing social stigma, which then results in lower mental health compared to the general population [3,4,5,6]. The leading framework for explaining disparities in mental (and sexual) health and suicide based on sexual minority status is the minority stress model [7,8]. The model, as applied to sexual minorities, recognizes the effects of distal stressors (i.e., episodes of overt discrimination or negative events that are externally located) and proximal stressors (i.e., concealment, expectation of rejection, and internalized homophobia that are internal processes) in determining mental and sexual health [9,10,11,12]. Existing research has investigated the unique role of these stressors on sexuality and SQoL for sexual minority individuals, however, these often focus on gay and bisexual men [2].

Internalized homophobia was conceptualized prior to its inclusion in the minority stress model and represents the turning of negative social attitudes regarding homosexuality against the self [13]. Internalized homophobia is further characterized by an intrapsychic conflict between experiencing affection or desire for someone of the same gender and feeling a need to be heterosexual [14]. Because the term may inaccurately suggest a phobic reaction, researchers have used alternative terms such as internalized heterosexism, sexual prejudice, sexual stigma, and homonegativity to express the same concept. When possible, we prefer the terms homonegativity, binegativity [15], or homo-bi-negativity. Research has shown that internalized homonegativity can be considered a distinct construct among the minority stressors [16], and that it has a negative impact on the mental health and well-being of LGB people [17,18,19]. Previous research on gay men has identified internalized homonegativity as a significant barrier to self-acceptance and the development of a positive sexual identity [20]. In addition, the meta-analytic review conducted by Newcomb and Mustanski [21] demonstrated the consistency of research linking internalized homonegativity to mental health problems. Similar results have been found for binegativity, which can result in higher levels of psychological distress and lower levels of life satisfaction [22].

Internalized homonegativity is a proximal stressor (related to intrapsychic factors), but is also strongly connected to the negative social stigma of the environment in which the person lives [16,23]. Italy is a notable case compared to other southern European countries (such as France and Spain) in terms of a lack of legal recognition of LGBTQIA+ rights. Italy was, in 2016, the last West European country to adopt a civil partnership law, and it has not yet adopted any law to combat hate crimes and hate speech motivated by homo-bi-transphobia [24]. Therefore, we can expect homo-bi-negativity to be a relevant stressor. In a previous Italian study, furthermore, bisexual people reported feeling prejudiced and discriminated against by both heterosexual and homosexual people [25]. This may be a factor leading to the concealment of their sexual orientation, which is a stressor that also has been framed within the minority stress model [15]. Scandurra et al. [12] investigated the role of minority stressors in a sample of Italian men and women, and found that internalized binegativity was positively associated with psychological distress. A study by Grabski and colleagues [2] on gay and bisexual men in Poland—which is also a country with increasing rates of violence against LGBTQIA+ people and a lack of protective legislation—showed that minority stressors, in particular internalized homonegativity, are significant independent correlates of sexual quality of life.

Regarding women who have sex with women (WSW), internalized homonegativity has been associated with poor relationship quality [26] and sexual satisfaction [10]. Internalized homonegativity consistently interferes with the individual’s psychological well-being [21] and can affect emotional intimacy [27] and the quality of romantic and sexual relationships [16,28]. We base our research question on the minority stress model, focusing specifically on the specific role that internalized homo-bi-negativity plays in the SQoL in Italian lesbian and bisexual cisgender women. To date, no research has examined SQoL and internalized homo-bi-negativity among Italian sexual minority women. We will compare lesbians and bisexuals in their levels of internalized homo-bi-negativity and in their SQoL. Our hypotheses are as follows:

(1) Regarding homo-bi-negativity, in line with previous literature [29], we hypothesize that bisexual women may have higher levels of internalized homo-bi-negativity when compared to lesbian women.

(2) Regarding SQoL, two comparative studies [10,30] found no differences in overall sexual satisfaction between bisexual and lesbian women. We hypothesize that we will find the same result for SQoL.

(3) We will examine the relationship between internalized homo-bi-negativity and SQoL, hypothesizing that this will show a negative association.

## 2. Materials and Methods

### 2.1. Participants

A quantitative methodology based on an online questionnaire was used. The data were collected from February to June 2022. The recruitment process entailed identifying groups on various social media platforms (e.g., Facebook and Instagram) whose members might be interested in participating in studies on topics related to being lesbian or bisexual. The selected individuals and organizations were sent a message introducing the research plan and the content of the questionnaire, along with a link to access the questionnaire. They were asked to share the link on their social media pages and with people who might be available and interested. The inclusion criteria for the study were being a woman, being 18 years of age or older, and having had sexual contact with another woman. We used the same questionnaire to collect data for another project targeting the same population, where we included closed-ended questions about experiences with different sexual practices and their prevalence, and these data were analyzed separately [31].

The study was approved by the University Ethics Committee of the University of Milan Bicocca. Participants were provided with a brief description of the content and objectives of the project, as well as the ethical guidelines and privacy policy. Informed consent was obtained from all participants. The final sample included 686 women.

### 2.2. Measures and Procedure

Internalized homonegativity was assessed using the Lesbian Internalized Homophobia Scale [32,33]. The Italian version of the scale comprised twelve items on five different dimensions of internalized homonegativity relevant for lesbian/bisexual women. These included connection to the lesbian community, public identification as lesbian, personal feelings about being lesbian, moral and religious attitudes towards lesbianism, and attitudes towards other lesbians. We adapted all the items to refer to lesbian and bisexual identities. The 12 items are scored on a 7-point Likert scale, ranging from 1 (strongly disagree) to 7 (strongly agree). The total score, which ranges from 7 to 84, is obtained from the sum of the 12 items, and a higher total score indicates higher levels of internalized homo-bi-negativity. Sample items are “I frequently make negative comments about other lesbian or bisexual women” and “I feel isolated and separated from other lesbian and bisexual women”. In the present study, the scale showed good internal consistency (Cronbach’s α = 0.74).

Sexual Quality of Life was assessed through the Female Sexual Quality of Life Questionnaire (F-SQoL) [34]. The areas assessed include emotions, sexuality, feelings of worthlessness, and repressing one’s own emotions. The scale included 18 questions, with responses ranging from 0 (strongly disagree) to 5 (strongly agree). All items except the positive items 1, 5, 9, 13, and 18 were reverse scored. The total score, ranging from 0 to 6, is obtained by averaging the 18 items, and a higher total score indicates a higher SQoL. Sample items are “When I think about my sexual life, I feel frustrated” and “When I think about my sexual life, it is an enjoyable part of my life”. In the present study, the scale showed good internal consistency (Cronbach’s α = 0.88).

Self-Designed Questionnaire: a questionnaire with single and multiple choice questions was developed for this data collection. The information collected through this questionnaire, and used in the present analysis, includes demographic data (i.e., age, education), gender identity, sexual and romantic attraction, and the label that best describes sexual orientation via the following question: “Despite the narrow definition, we ask you to select the label that best describes you”, with a multiple choice answer (lesbian woman, bisexual woman, heterosexual woman).

### 2.3. Statistical Analyses

Statistical analyses were performed with SPSS, version 29. *t*-tests were employed to perform the group comparisons, while multiple linear regression was employed to assess the joint influence of predictors on the outcome variable. Due to the unbalanced sample size, we used Welch’s *t* test, which is robust for unequal sample size and unequal variances. The linear regression model included a dichotomous sexual orientation variable (lesbian or bisexual) and internalized homo-bi-negativity as predictors. To show that the association between the predictors and sexual satisfaction was not due to the indirect effects of age, we controlled for age by including it in the model.

## 3. Results

### 3.1. Demographic Data

Of the 989 responses collected, 723 (73.1%) were complete. Of the complete responses, 37 women (5.12%) identified as heterosexual and had had sexual encounters with other women. Although heterosexual women may belong to the WSW group, since they may have a heterosexual identity but still engage in sexual behaviors or feel sexual attraction towards other women [35,36], we excluded them from our analysis. The final sample comprises a total of 686 women, with 31.63% (N = 217) identifying as lesbian and 68.37% (N = 469) as bisexual. The mean age of lesbian participants is 31.05 years (*SD* = 8.24), while bisexual individuals have a slightly lower mean age of 28.16 years (*SD* = 6.07). A comparison performed using Welch’s *t*-test shows that the bisexual respondents are statistically younger than the lesbian respondents (*t*(328.55) = 4.61, *p* < 0.001). Further comparisons performed using Welch’s *t*-test show that, in our sample, bisexual respondents have, on average, a higher education level (*t*(364.22) = −2.29, *p* = 0.02) and that more bisexual respondents are in non-monogamous (as opposed to monogamous) relationships compared to lesbian respondents (*t*(554.11) = −6.86, *p* < 0.001). Further sociodemographic data are outlined in Table 1.

### 3.2. t-Tests

An independent sample Welch’s t-test was performed to test for differences in internalized homo-bi-negativity and SQoL in lesbian and bisexual women. Results indicated a significant difference in mean internalized homo-bi-negativity (*t*(465.41) = −2.92, *p* = 0.004). Lesbian women reported a mean of 25.30 (*SD* = 8.07, range 12–55), whereas bisexual women reported a higher mean of 27.31 (*SD* = 9.01, range 12–66), hence supporting hypothesis 1 (see Table 2). Results showed a significant difference also in the mean SQoL between the groups (*t*(474.77) = 3.88, *p* < 0.001). Lesbian women reported a mean sexual quality of life of 4.71 (*SD* = 0.72, range 2–6), while bisexual women had a lower mean of 4.47 (*SD* = 0.82, range 2–6), hence disconfirming hypothesis 2. The 95% confidence intervals for both tests excluded 0, supporting the observed differences. Statistical significance was assessed at the level of 0.05.

### 3.3. Linear Regression

The multiple linear regression model was employed to examine the predictors of SQoL, taking into account age, internalized homo-bi-negativity (LIHS), and sexual orientation. The independent variables significantly predicted SQoL, *F* (3, 682) = 10.246, *p* < 0.001. Both internalized homo-bi-negativity (*p* < 0.001) and the dichotomous sexual orientation variable (*p* = 0.002) showed significant associations with SQoL. The coefficient for the LIHS was −0.01 (*SE* = 0.00, *t* = −4.08, *p* < 0.001). The statistically significant negative association suggests that, as indicated in hypothesis 3, as internalized homo-bi-negativity increases, the sexual quality of life decreases. The means of the sexual orientation variable allow for a nuanced examination of differences between lesbian and bisexual women (see Table 3). Bisexual individuals, coded as 2 in the sexual orientation variable (with lesbians coded as 1), exhibited an average decrease in SQoL of 0.21-unit compared to their lesbian counterparts (*SE* = 0.07, *t* = −3.15, *p* = 0.002). Age did not emerge as a statistically significant predictor (*p* = 0.722). The R-squared value was found to be 0.043, suggesting that the included predictors collectively explained 4.3% of the variability in the SQoL scores. It is essential to acknowledge that a substantial portion of the variability remains unexplained, highlighting the multifaceted nature of SQoL.

## 4. Discussion

The primary aim of our study was to compare levels of internalized homo-bi-negativity and sexual quality of life (SQoL) between lesbian and bisexual women. Additionally, we aimed to examine the influence of internalized homo-bi-negativity on SQoL. In line with our first hypothesis, we observed higher levels of internalized homo-bi-negativity among bisexual women. We did not expect differences in SQoL, but, contrary to our second hypothesis, lesbian women showed higher levels of SQoL. Our linear regression model confirmed the role of internalized homo-bi-negativity in predicting lower SQoL. However, it is important to acknowledge that a significant portion of the variance remains unexplained, highlighting the need to identify additional relevant variables.

Our first hypothesis that bisexual women may have higher levels of internalized homo-bi-negativity was confirmed. Since we frame internalized homo-bi-negativity as a proximal and intrapsychic stressor that is also strongly related to social stigma, the lack of the social representation of bisexuality may help explain the internalization of negative feelings and beliefs about oneself [6]. This finding can be explained by the fact that bisexual individuals are subject to “double discrimination”, in that they receive negative attitudes from both the heterosexual majority and from lesbian and gay people [25,37]. Although bisexual identities are becoming more visible (and more women are identifying as bisexual), there are few organized communities and activist spaces for bisexuality when compared to other minority groups. Hayfield et al. [38] found that bisexual women do not feel welcome in LGBTQIA+ communities. This isolation can lead to less opportunities of affiliation, and may lead to a greater internalization of a negative image of oneself and one’s sexuality. Bisexual people may experience a “double closet”, where they have to conceal their same-sex attraction from their heterosexual peers while hiding their heterosexual attraction from their gay and lesbian peers [6,39,40].

Our second hypothesis that there would be no differences in the overall SQoL between bisexual women and lesbians was not supported. Lesbian women, on average, had a better SQoL. Although previous studies did not find differences in sexual satisfaction between these groups [10,30], SQoL measures different aspects of sexuality and is more connected to well-being. Previous research has found that bisexual women are more likely than lesbians to experience frequent psychological distress and poor general health [41]. They have a higher risk of self-harm compared to lesbian women [42]. A systematic review and meta-analysis conducted by Ross et al. [43] found elevated rates of depression and anxiety for bisexuals when compared to heterosexuals, gays, and lesbians. Regarding sexuality-related outcomes, bisexual men have been found to be at risk for worse scores [44]. We are not aware of any other studies using the SQoL-F measure conducted on a group of sexually active lesbian and bisexual women that would allow us to make accurate comparisons. Regarding homosexual and bisexual men, Grabski et al. [2] used the male version of the SQoL scale and found no differences in the quality of sexual life among gay and bisexual Polish men. Our results point to worse outcomes (lower sexual quality of life and higher internalized homo-bi-negativity) for bisexual women.

The linear regression indicated that higher levels of homo-bi-negativity predicted a lower SQoL, which was consistent with our third hypothesis. The model also confirmed what was addressed in the t-test, that women with a bisexual sexual orientation had lower levels of SQoL. Regarding lesbian and bisexual women, we are not aware of any studies that have used SQoL as an outcome. Similar studies have been led assessing the impact of homonegativity or binegativity on sexual satisfaction, but have shown inconsistent results. Kuyper and Vanwesenbeeck [10] found that internalized homonegativity was associated with lower levels of sexual satisfaction and higher levels of sexual dysfunction among lesbian women. Internalized homonegativity leads lesbian and bisexual individuals to redirect negative social evaluations onto themselves, which may affect sexuality through the cognitive dissonance caused by the conflict between one’s sexual activity and the belief that such activity is not legitimate [2]. This finding can be explained by the fact that internalized homo-bi-negativity is constantly in operation. Bisexual individuals receive more blame than other sexual orientations [45], which may lead them to internalize blame and negative attitude toward bisexuals themselves. Therefore, internalized binegativity is related to the social system, but also highly personal and can be a stressor that impacts intimate settings, such as sexuality [2,28].

However, other studies have not found this association. For example, Mark and Coll. [46] led a study on a sample of bisexual women, and they found that internalized binegativity did not negatively predict sexual or relationship satisfaction. Shepler et al. [47] found a similar result when assessing sexual satisfaction among LGB adults. In their study, internalized homonegativity did not contribute to sexual satisfaction. This finding may be due to other variables contributing to sexual satisfaction, as exemplified by the findings of Henderson et al. [48]. The authors found that internalized homonegativity and sexual satisfaction were significantly and negatively correlated. However, when including other variables in the model, the association between the two was no longer significant [48]. In our model, with SQoL being the outcome variable, we observed the hypothesized effect, but the majority of the variance remains unexplained, calling for future studies to further disentangle the role uniquely played by other factors.

## 5. Conclusions

A strength of the present study is that it differentiates between monosexual and non-monosexual orientations, rather than considering lesbian and bisexual women as a monolithic group [49], as our results show that sexual orientation is a significant factor influencing sexual quality of life. Another strength of the methodology of this study is the use of SQoL-F as an outcome variable. None of the items in the scale mention specific sexual practices, and the questions that refer to partnered sex do not mention or assume the gender of the partner, making it a useful scale with a population of WSW. The scale does not mention orgasm or sexual functioning in general. In clinical and research practice, the quality of one’s sexuality is often conflated with the presence of orgasms or sexual function. We argue that the focus should shift toward sexual quality of life, as the subjective experience of well-being in one’s sexuality, and away from heterocentric measures of sexual well-being. In general, research on sexual minorities should move away from the function–dysfunction dichotomy, and instead promote an affirming and resilience-oriented perspective.

The study is not without its limitations. As with any cross-sectional study, causal claims cannot be made from the data. Although our study suggests that internalized homo-bi-negativity is associated with SQoL in lesbian and bisexual women, it was not designed to assess the full range of factors that may be involved or even how such factors may interact with one another. It is crucial to consider many other factors, such as general health and the current relationship, if present (e.g., satisfaction, quality, type, the number or genders of the partners). Another limitation is the lack of assessment focusing on ethnicity or other sociodemographic variables that may be relevant to an individual’s sexuality. Regarding internalized homo-bi-negativity, we chose a measure that had been validated on an Italian population of lesbians. We adapted all items so that they included bisexual women. Future studies could validate the scale on a bisexual population, or compare if the two groups have distinct characteristics. Finally, we assessed the participants only through self-report questionnaires. This may protect the person’s privacy, but it may also result in self-serving and social-desirability biases, as well as random completion.

Future research should examine factors that promote mental and sexual well-being in bisexual women. Future studies should examine factors other than sexual orientation and internalized homo-bi-negativity that help explain SQoL in sexual minority women, including other minority stressors or socio demographic characteristics, such as educational level, relational orientation, or relational status. Furthermore, the minority stress model, which is the theoretical framework used in the present work, has been criticized and is constantly updated. Future research may try to test rival hypotheses for explaining worse health outcomes in LGBTQIA+ persons [50].

The findings hold crucial implications for clinical practice with sexual minority women, particularly in the Italian context. The observed higher levels of internalized homo-bi-negativity among bisexual women underscore the importance of acknowledging and addressing the unique challenges faced by this group. Clinicians should be attuned to the double discrimination that bisexual individuals may experience, both from the heterosexual majority and from within LGBTQIA+ communities. Disparities in SQoL between lesbian and bisexual women emphasize the need for tailored interventions that take into account the diverse experiences within the sexual minority spectrum [51]. Addressing internalized homo-binegativity may be relevant for the well-being of both lesbian and bisexual women, as it may act as a barrier to self-compassion [52,53]. Clinicians should employ affirmative approaches that contextualize internalized homo-bi-negativity, fostering self-compassion and resilience. Moreover, the study advocates for a shift in clinical and research perspectives, moving away from the traditional function–dysfunction dichotomy and toward a more holistic and affirming stance that considers the multifaceted nature of sexual well-being among sexual minority women. Clinicians should receive education in the area of sexuality, regardless of their discipline, so that they can engage in conversation and promote affirming behaviors. Our findings underscore the need for tailored clinical interventions that address the specific stressors faced by bisexual individuals, recognizing the impact of societal and intra-community discrimination, as well as double discrimination. Furthermore, when bisexual people are part of other minorities (e.g., ethnicity, religion, living with a chronic illness or a disability, etc.), these oppressions may add to the double discrimination, and may be even more socially marginalized. Future research and clinical programs could focus on the bisexual experience intersectionally in Italy.

In conclusion, this study sheds light on the nuanced dynamics of internalized homo-bi-negativity and SQoL among Italian lesbian and bisexual cisgender women. Our research is particularly relevant, given the mediatic, political, and governmental situation in Italy, which is highly invalidating for LGBTQIA+ subjectivities. The study advocates for a shift towards more inclusive and affirming approaches in both clinical and research settings, acknowledging the complexity of factors that influence the well-being of sexual minority women. Current school sex education curricula could benefit from LGBTQIA+ competent information, ensuring that topics such as minority stressors can be addressed.

## Figures and Tables

**Table 1 healthcare-12-00638-t001:** Descriptive statistics of the socio-demographic characteristics of the sample.

Variable		Lesbian 31.63% (N = 217)	Bisexual 68.37% (N = 469)
Age (Mean–SD)		(31.05–8.24)	(28.16–6.07)
Age (range)		(18–70)	(18–59)
Education level	Elementary School	0% (0)	0% (0)
% (N)	Middle School	1.38% (3)	1.07% (5)
	Professional Diploma	4.61% (10)	1.07% (5)
	High school diploma	34.56% (75)	30.70% (144)
	University degree or Postgraduate education	57.60% (125)	64.82% (304)
	Other	1.84% (4)	1.92% (9)
	I prefer not to answer	0% (0)	0.43% (2)
Self-label % (N)	Lesbian vs. bisexual	31.5% (217)	68.4% (469)
Relationship Status % (N)	Monogamous relationship	65.44% (142)	47.97% (225)
	Consensual non-monogamy (open relationship, polyamory, etc.)	7.37% (16)	20.04% (94)
	Non-consensual non-monogamy relationship	0.92% (2)	2.56% (12)
	Single (monogamous)	17.97% (39)	13.65% (64)
	Single (consensual non-monogamous)	3.69% (8)	10.23% (48)
	Not interested in dating someone	2.76% (6)	2.13% (10)
	Other	1.84% (4)	3.41% (16)

**Table 2 healthcare-12-00638-t002:** Comparison of sexual quality of life and internalized homo-bi-negativity between lesbian and bisexual women.

Variable	*t*-Value	df	*p*-Value	Standard Error	CI (Lower, Upper)	Hedges’g	Mean Score Lesbian	Mean Score Bisexual
Sexual Quality of Life	3.88	474.77	<0.001	0.06	(0.12, 0.36)	0.303	4.71 (*SD* = 0.72)	4.47 (*SD* = 0.82)
Internalized Homo-bi- negativity	−2.92	465.41	0.004	0.69	(−3.36, −0.66)	0.312	M = 25.30, (*SD* = 8.07)	M = 27.31, (*SD* = 9.01)

Note: Results of independent sample Welch’s *t*-tests for the comparison between groups (Lesbian vs. Bisexual) for the mean scores of SQL and internalized homo-bi-negativity. The tests were conducted on a sample of 217 lesbian women and 469 bisexual women. The table includes Welch’s *t* test-values, degrees of freedom, *p*-values, standard error, 95% confidence intervals, effect size measured through Hedges’ g, and mean scores for each group.

**Table 3 healthcare-12-00638-t003:** Multiple linear regression analysis of factors influencing sexual quality of life in lesbian and bisexual women.

Predictor	Coefficient	Standard Error	*p*-Value	Mean Lesbian	Mean Bisxual
(Intercept)	5.22	0.20	<0.001		
Age	0.00	0.00	0.72	good internal consistency (Cronbac 31.04	28.16
Internalized Homonegativity	−0.01 ***	0.00	<0.001	4.71	4.47
Sex Orientation	−0.21 **	0.7	0.002	25.30	27.31

Note: Results of a multiple regression analysis examining the relationship between selected predictors and the Sexual Quality of Life (SQoL) in lesbian and bisexual women. The analysis involved 686 participants. All predictors’ coefficients are standardized. R^2^ = 0.043, R^2^ adj = 0.039. *F* (3, 682) = 10.25, *p* < 0.001; ** *p* < 0.01; *** *p* < 0.001.

## Data Availability

The data used for the study are openly available online at https://github.com/solstice10/WSW/tree/SQoL.

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
