# Peer review of "Internalized Homonegativity and Sexual Quality of Life in Italian Lesbian and Bisexual Women"

_healthcare, 2024, doi:10.3390/healthcare12060638_

Round 1

Reviewer 1 Report

Comments and Suggestions for Authors

First of all, I would like to congratulate the authors of this paper for tackling this very interesting topic. Overall I think the research is well constructed. This study addresses the relationship between sexual quality of life and internalised homonegativity among cisgender lesbian and bisexual women, for the case of Italy. It is based on the minority stress model.

The following are some comments on the different parts of the paper.

Abstract: it is clear, the objective of the research, the methodology and the main findings are defined. I recommend that when explaining the methodology, the authors first specify that it is quantitative and then the online questionnaire.

Introduction: This section is clear and coherent. It presents a brief context about the object of study and makes a good review of the literature to build the hypotheses. However, I think there is one aspect that could be improved: it is said that Italy is a remarkable case in comparison with other southern European countries in terms of lack of recognition of the indicated collective. However, I think it is necessary to detail at least which countries are referred to (Spain for example) and in which concrete terms (mention some).

The hypotheses set out are clear and well constructed through the brief review of studies conducted.

Materials and methods: in general terms this section is well constructed but, in my opinion, some things need to be specified.

- In this section I underline the need to go from the more general to the more specific: a quantitative methodology based on a questionnaire was used. From this point onwards, explain how the information was collected (how the questionnaire was passed on to the people).

- The limitations of conducting a questionnaire online (randomness, probability...) should be briefly outlined. The use of this way of completing a questionnaire is widely accepted in the scientific community, but I think that possible limitations should be established.

- It is necessary to indicate the final sample (number of participants). It is indicated at the beginning of the results but it is necessary for potential readers that it is present in this section.

The statistical analysis seems accurate given the type of data obtained. The use of the t-test to compare groups is very appropriate. Likewise, it is also appreciated that the need for the Welch's t-test is specified due to the sample size..

Results: it is necessary to highlight that the regression used is multiple linear regression (only linear). The model can help to understand this phenomenon although the predictive power is limited by looking at the R values.

Discussion: is consistent with the results obtained. The hypotheses are validated or refuted and related to preliminary studies.

Conclusions: these are clear and straightforward. It would be good to include in this section the limitations of the study (these can be found in the previous section) as well as to establish what possible future lines of research its findings allow for.

Author Response

First of all, I would like to congratulate the authors of this paper for tackling this very interesting topic. Overall I think the research is well constructed. This study addresses the relationship between sexual quality of life and internalised homonegativity among cisgender lesbian and bisexual women, for the case of Italy. It is based on the minority stress model.
>> We thank the reviewer for their interest in the topic of our manuscript and for the feedback provided, which we integrated in the current version of the manuscript.

The following are some comments on the different parts of the paper.

Abstract: it is clear, the objective of the research, the methodology and the main findings are defined. I recommend that when explaining the methodology, the authors first specify that it is quantitative and then the online questionnaire.
>> In the current version of the abstract we make clear mention of the quantitative methodology used.

Introduction: This section is clear and coherent. It presents a brief context about the object of study and makes a good review of the literature to build the hypotheses. However, I think there is one aspect that could be improved: it is said that Italy is a remarkable case in comparison with other southern European countries in terms of lack of recognition of the indicated collective. However, I think it is necessary to detail at least which countries are referred to (Spain for example) and in which concrete terms (mention some).

>> We thank the reviewer for their hint on adding a comparison: they will now find a reference to a paper that considers the case of Italy as well as France and Spain in terms of legal recognition of the LGBTQ+ community. Regarding the concrete example, we opted for mentioning the recognition of same sex countries, which Italy adopted much later than its neighbor countries.

The hypotheses set out are clear and well constructed through the brief review of studies conducted.

Materials and methods: in general terms this section is well constructed but, in my opinion, some things need to be specified.

- In this section I underline the need to go from the more general to the more specific: a quantitative methodology based on a questionnaire was used. From this point onwards, explain how the information was collected (how the questionnaire was passed on to the people).

>> In line with the reviewer's suggestion, we specify at the beginning of the method section that a quantitative methodology based on an online questionnaire was used.

- The limitations of conducting a questionnaire online (randomness, probability...) should be briefly outlined. The use of this way of completing a questionnaire is widely accepted in the scientific community, but I think that possible limitations should be established.

>> We thank the reviewer for the suggestion, and we include the limitation of using a self-report questionnaire in the limitation section at the end of the manuscript.

- It is necessary to indicate the final sample (number of participants). It is indicated at the beginning of the results but it is necessary for potential readers that it is present in this section.

>> We add a mention of the final sample size at the end of the Participants section.

The statistical analysis seems accurate given the type of data obtained. The use of the t-test to compare groups is very appropriate. Likewise, it is also appreciated that the need for the Welch's t-test is specified due to the sample size..

Results: it is necessary to highlight that the regression used is multiple linear regression (only linear). The model can help to understand this phenomenon although the predictive power is limited by looking at the R values.

>>We thank the reviewer for the suggestion. We made sure that we mention that it is a linear regression throughout the manuscript.

Discussion: is consistent with the results obtained. The hypotheses are validated or refuted and related to preliminary studies.

Conclusions: these are clear and straightforward. It would be good to include in this section the limitations of the study (these can be found in the previous section) as well as to establish what possible future lines of research its findings allow for.

>> The readers can find limitations and future research in the conclusion paragraph.

Reviewer 2 Report

Comments and Suggestions for Authors

3 February 2024

The authors have written about a topic that would certainly fascinate readers, and to the credit of the authors, the paper does not attempt to wander into claims that fall beyond the ambit of the evidence available (cf. lines 293~294: "The study is not without limitation [. . .] causal claims cannot be made from the data").

In certain parts of the paper, the authors could have chosen to elaborate a bit further on certain claims or commentaries on methodology and results. In the "measures and procedure" section (2.2), the authors would have done well to explain reasons behind the rejection of certain forms of measurement and certain procedural steps; the measures and procedure section, in other words, could have offered an opportunity for the authors to explain the unique ways in which this submission methodologically differs from (or converge with) publications on vaguely similar or very similar topics. In lines 148~150, the authors also mention a curious phenomenon of the female respondents who self-identified as heterosexual---and yet had sexual encounters with other women. Although this group constituted a mere five percent of the responses, the authors might have tried to explain (even if briefly) the self-contradictory nature of this particular group of respondents (according to common sense, one cannot self-identify as a heterosexual while also having homosexual sexual activity. In lines 231~232, the authors also write, "Previous research has found that bisexual women are more likely than lesbians to experience frequent psychological distress and poor general health," but perhaps the authors could have provided more background information on the possible reason or reasons behind this phenomenon. The authors might have also considered lengthening the conclusion. In its present form, the conclusion seems almost more like some token summaries merely inserted into the paper (lines 316~324).

Author Response

The authors have written about a topic that would certainly fascinate readers, and to the credit of the authors, the paper does not attempt to wander into claims that fall beyond the ambit of the evidence available (cf. lines 293~294: "The study is not without limitation [. . .] causal claims cannot be made from the data").

>> The authors thank the reviewer for their feedback.

In certain parts of the paper, the authors could have chosen to elaborate a bit further on certain claims or commentaries on methodology and results. In the "measures and procedure" section (2.2), the authors would have done well to explain reasons behind the rejection of certain forms of measurement and certain procedural steps; the measures and procedure section, in other words, could have offered an opportunity for the authors to explain the unique ways in which this submission methodologically differs from (or converge with) publications on vaguely similar or very similar topics.

>>The rationale for chosing F-SQoL is outlined extensively in the Discussion section. We added a reflection on the choice of our scale for homo-bi-negativity, as well as some consideration on using self-reports. In our opinion these considerations are more suited for the discussion section, limiting the “Measures and procedure" section to the description of the scales that we used, but we are open to change it if necessary. 

 In lines 148~150, the authors also mention a curious phenomenon of the female respondents who self-identified as heterosexual---and yet had sexual encounters with other women. Although this group constituted a mere five percent of the responses, the authors might have tried to explain (even if briefly) the self-contradictory nature of this particular group of respondents (according to common sense, one cannot self-identify as a heterosexual while also having homosexual sexual activity. 

>> In literature, as well as activism, there is a rise of awareness regarding how one’s sexual attraction, identity, and one’s behavior do not necessarily conflate. Therefore, we argue it is plausible that a woman with a heterosexual identity may feel attraction towards other women, or engage in sexual relationships with other women. In line with the reviewer's suggestion, we elaborate on this in the manuscript, when discussing the participants, by including two references to previous studies.

In lines 231~232, the authors also write, "Previous research has found that bisexual women are more likely than lesbians to experience frequent psychological distress and poor general health," but perhaps the authors could have provided more background information on the possible reason or reasons behind this phenomenon. 

>> In line with another reviewer’s feedback, we expanded on the role of double discrimination, from both heterosexual and LG communities, for bisexual individuals, hoping that this will add context for our statement in the discussions.

The authors might have also considered lengthening the conclusion. In its present form, the conclusion seems almost more like some token summaries merely inserted into the paper (lines 316~324).

>>We thank the reviewer for the suggestion. In line with their and other reviewer’s feedback we moved the strength and limitation section, and we elaborated more on future research in the conclusion.

Reviewer 3 Report

Comments and Suggestions for Authors

This is an interesting project. The data here is rich. This paper could be strengthened by more clearly highlighting your research question(s) and clarifying the importance of your findings. 

In the introduction authors should better specify the specific situation lived by bisexual people about "double stigma" (also emerged in previous research on this topic conducted in Italy, as in case of 10.1108/IJSSP-05-2020-0157).

I would also have liked more social, political, and historical background on LGBT+ rights in Italy. Why Italy? What about Italy makes this case important?

About references, Trenton Mize's work on sexuality and discrimination might be of use. In addition, I would like to see more literature on the intersection of gender, race, and sexuality - also a discussion on the term/identity "bisexual" which has a contentious history regarding sex vs gender. 

In the conclusion part, I think more work needs to be done to highlight what contributions and significance this study offers, identifying further implications for practice and policy.

Author Response

This is an interesting project. The data here is rich. This paper could be strengthened by more clearly highlighting your research question(s) and clarifying the importance of your findings. 

>>We thank the reviewer for their positive feedback and for the suggestions, which we addressed in the current version of the paper.

In the introduction authors should better specify the specific situation lived by bisexual people about "double stigma" (also emerged in previous research on this topic conducted in Italy, as in case of 10.1108/IJSSP-05-2020-0157).

>>We thank the reviewer for recommending this work. It is now included in the introduction section together with a mention of the double stigma, which is discussed more extensively in the discussion.

I would also have liked more social, political, and historical background on LGBT+ rights in Italy. Why Italy? What about Italy makes this case important?

>>Italy is the country where we as researchers and clinicians are based, and therefore we are very aware of the cultural panorama and the. Furthermore, Italy is remarkably behind compared to its neighboring countries. The readers can find a mention of this, especially regarding couple’s rights in the current version of the manuscript in the introduction section.

About references, Trenton Mize's work on sexuality and discrimination might be of use. In addition, I would like to see more literature on the intersection of gender, race, and sexuality - also a discussion on the term/identity "bisexual" which has a contentious history regarding sex vs gender. 

>> The reader can find a mention of the importance of considering intersections especially when considering the bisexual population, which is already subject to double discrimination. However, these interactions are factors that we did not assess. We argue future studies should include them especially in relation to bisexuality.

In the conclusion part, I think more work needs to be done to highlight what contributions and significance this study offers, identifying further implications for practice and policy.

>> In line with the reviewer’s suggestion we elaborate further on the implication (e.g. for sex education programs and on clinical implications).

Round 2

Reviewer 2 Report

Comments and Suggestions for Authors

The two authors have addressed this reviewer's critiques/commentaries on the first draft.

Author Response

The two authors have addressed this reviewer's critiques/commentaries on the first draft.

>>We thank the reviewer for checking the manuscript in its new form.